# The Necessary and Sufficient Conditions When Global and Local Fidelities Are Equal

**DOI:** 10.3390/e25071093

**Published:** 2023-07-21

**Authors:** Seong-Kun Kim, Yonghae Lee

**Affiliations:** Department of Liberal Studies, Kangwon National University, Samcheok 25913, Republic of Korea; kimseong@kangwon.ac.kr

**Keywords:** quantum information, quantum fidelity

## Abstract

In the field of quantum information theory, the concept of quantum fidelity is employed to quantify the similarity between two quantum states. It has been observed that the fidelity between two states describing a bipartite quantum system A⊗B is always less than or equal to the quantum fidelity between the states in subsystem *A* alone. While this fidelity inequality is well understood, determining the conditions under which the inequality becomes an equality remains an open question. In this paper, we present the necessary and sufficient conditions for the equality of fidelities between a bipartite system A⊗B and subsystem *A*, considering pure quantum states. Moreover, we provide explicit representations of quantum states that satisfy the fidelity equality, based on our derived results.

## 1. Introduction

Quantum fidelity [1,2] is a fundamental and indispensable tool in quantum information theory for quantifying the closeness between two quantum states that describe a quantum system. Among its various applications, quantum fidelity plays a crucial role in evaluating the success of key quantum communication tasks within quantum Shannon theory, including quantum teleportation [3], quantum state merging [4,5], and quantum state redistribution [6,7]. To illustrate the importance of quantum fidelity, we focus on the task of quantum state merging. In this task, two users, Alice and Bob, initially possess separate parts *A* and *B* of a shared quantum state $\rho^{AB}$. By employing local operations and classical communication assisted by shared entanglement, their objective is to merge Alice’s quantum state with Bob’s, resulting in the target state $\rho^{B^{\prime }B}$, where $B^{\prime }$ corresponds to Bob’s quantum system. Upon completion of the merging process, how can they ascertain the closeness of the resulting state to the desired target state? Without the aid of the quantum fidelity, it would be impossible to compare and assess the similarity between these states.

In this study, we consider the following inequality [8]:(1)$$F\left(\rho^{AB},\sigma^{AB}\right)\leq F\left(\rho^{A},\sigma^{A}\right),$$
where $\rho^{AB}$ and $\sigma^{AB}$ represent the quantum states of the bipartite system AB, and $\rho^{A}$ and $\sigma^{A}$ represent the reduced states of $\rho^{AB}$ and $\sigma^{AB}$ corresponding to the quantum system *A*. This inequality demonstrates that for any given pair of bipartite quantum states, the quantum fidelity on the bipartite quantum system AB is always less than or equal to the quantum fidelity on the local quantum systems *A*. To provide a simple illustration, let us examine the scenario of two EPR pairs [9]:(2)$${|\phi^{\pm }\rangle }^{AB}=\frac{1}{\sqrt{2}}\left({|00\rangle }^{AB}\pm {|11\rangle }^{AB}\right),$$
where $|0\rangle$ and $|1\rangle$ are the computational basis of a two-dimensional quantum system. In this context, the quantum fidelity between $\phi^{+}$ and $\phi^{-}$ is found to be zero. However, when we evaluate their fidelity on the local quantum system *A*, it becomes one. This intriguing observation implies that the quantum states $\phi^{+}$ and $\phi^{-}$ are indistinguishable on the local quantum system *A*, indicating complete identity. However, on the bipartite quantum system AB, they exhibit complete distinctness.

The inequality Equation (Equation 1) is easy to understand, as discussed earlier. However, determining the conditions under which the fidelities in Equation (Equation 1) become equal is difficult. This study focuses on overcoming this limitation by considering pure bipartite quantum states ${|\psi \rangle }^{AB}$ and ${|\phi \rangle }^{AB}$. We aim to investigate the conditions for fidelity inequality as stated in Equation (Equation 1) and provide explicit representations of pure bipartite quantum states that satisfy these conditions.

The remainder of this paper is organized as follows: In Section 2, we introduce the definitions of global and local fidelities, along with the assumptions and lemmas that form the foundation of our main results. Section 3 presents a comprehensive calculation of the global and local fidelities. In Section 4, we present the conditions that establish the equivalence for fidelity equality. Section 5 is devoted to presenting specific forms of pure bipartite quantum states that fulfill these equivalent conditions. Finally, in Section 6, we discuss our findings, their implications, and outline potential avenues for future research.

## 2. Definitions, Assumptions, and Lemmas

In this section, we provide the definitions, assumptions, and lemmas that are employed throughout this work.

To begin, we consider finite-dimensional Hilbert spaces H. The notation $H^{X}$ denotes a Hilbert space representing a quantum system *X*. The tensor product $H^{A}⊗H^{B}$ signifies a composite quantum system comprising two quantum systems *A* and *B*, which can be denoted as A⊗B or simply AB. The dimension of the Hilbert space $H^{X}$, denoted as dimX, corresponds to the dimension of the quantum system *X*.

Let D(H) denote the set of density operators on a Hilbert space H. In other words, D(H)={ρ∈L(H):ρ≥0,Tr[ρ]=1}, where L(H) denotes the set of all linear operators on H. The elements within D(H) are referred to as quantum states. If a quantum state ρ can be expressed as a rank-1 projector, i.e., it can be represented as
(3)$$\psi :=|\psi \rangle \langle \psi |,$$
where $|\psi \rangle$ is a normalized vector in the Hilbert space H, it is referred to as a pure state. Here, the unit vector $|\psi \rangle$ is also considered a pure quantum state. Quantum states that are not pure are referred to as mixed states, and they are denoted by ρ or σ in this paper.

The trace, Tr[ρ], of a quantum state ρ operating on a Hilbert space H is defined as
(4)$$\mathrm{Tr}\left[\rho \right]:=\sum_{j}\langle j|\rho |j\rangle ,$$
where $\left\{|j\rangle \right\}$ represents any orthonormal basis of the Hilbert space H. For a bipartite quantum state $\rho^{AB}$ on a Hilbert space $H^{A}⊗H^{B}$, the partial trace over the Hilbert space $H^{B}$ is defined as
(5)$${\mathrm{Tr}}_{B}\left[\rho^{AB}\right]:=\sum_{j}\left(I^{A}⊗{\langle j|}^{B}\right)\rho^{AB}\left(I^{A}⊗{|j\rangle }^{B}\right),$$
where $I^{A}$ denotes the identity matrix on the quantum system *A*, and $\left\{{|j\rangle }^{B}\right\}$ represents any orthonormal basis of the Hilbert space $H^{B}$. In this scenario, the quantum state $\rho^{A}:={\mathrm{Tr}}_{B}\left[\rho^{AB}\right]$ obtained on the Hilbert space $H^{A}$ is referred to as the reduced quantum state of $\rho^{AB}$.

In this study, we focus on investigating the quantum fidelity [8] between two quantum states ρ and σ that represent the same quantum system. The quantum fidelity is defined as
(6)$$F\left(\rho ,\sigma \right)={∥\sqrt{\rho }\sqrt{\sigma }∥}_{1}^{2}={\left(\mathrm{Tr}\sqrt{\sqrt{\rho }\sigma \sqrt{\rho }}\right)}^{2}.$$ In particular, when considering two pure quantum states $|\psi \rangle$ and $|\phi \rangle$, the quantum fidelity can be straightforwardly calculated as $F\left(\psi ,\phi \right)=|\langle \psi |\phi \rangle {|}^{2}$. We also investigate two pure quantum states ${|\psi \rangle }^{AB}$ and ${|\phi \rangle }^{AB}$ on the bipartite quantum system AB, and with the assumption that dimA=2 and dimB≥2. For convenience, we use the notations
(7)$$\begin{array}{ccc} F^{AB} & := & F({|\psi \rangle }^{AB},{|\phi \rangle }^{AB}), \end{array}$$
(8)$$\begin{array}{ccc} F^{A} & := & F(\rho_{\psi }^{A},\rho_{\phi }^{A}), \end{array}$$
where $\rho_{\psi }^{A}$ and $\rho_{\phi }^{A}$ represent the reduced quantum states of pure bipartite quantum states ${|\psi \rangle }^{AB}$ and ${|\phi \rangle }^{AB}$, respectively. When referring to the given quantum states ${|\psi \rangle }^{AB}$ and ${|\phi \rangle }^{AB}$, we use the terms $F^{AB}$ and $F^{A}$ to present the *global* fidelity and the *local* fidelity, respectively. Thus, the fidelity inequality in Equation (Equation 1) can be expressed as
(9)$$F^{AB}\leq F^{A}.$$

Finally, we introduce two lemmas that will be used in the subsequent sections.

**Lemma** **1.**
*For any two complex numbers α and β, we have*

(10)
Re(αβ*)=|αβ|⇒β=kα,


(11)
$$\begin{array}{c} |\alpha |-|\beta |=|\alpha -\beta |\Leftrightarrow \beta =p\alpha , \end{array}$$

*where $\beta^{*}$ denotes the complex conjugate of β, k is a real number, and p is a real and non-negative value.*


**Proof.** (i) Assume that Re(αβ*)=|αβ| holds for any two complex numbers α and β. Given that α and β are complex, they can be expressed as α=a+ib and β=c+id using some real numbers *a*, *b*, *c*, and *d*. Notably,
(12)Re(αβ*)=Re((a+ib)(c−id))=Re((ac+bd)+i(bc−ad))=ac+bd,
(13)$$\begin{array}{ccc} \left|\alpha \beta \right| & = & \left|\left(a+ib\right)\left(c+id\right)\right|=\left|\left(ac-bd\right)+i\left(bc+ad\right)\right|=\sqrt{{\left(ac-bd\right)}^{2}+{\left(bc+ad\right)}^{2}}. \end{array}$$ Consequently, the assumption implies that ${\left(ad-bc\right)}^{2}=0$; thus, ad=bc. Therefore,
(14)$$\beta =c+id=\frac{ad}{b}+id=\frac{d}{b}\left(a+ib\right)=k\alpha ,$$
where k=d/b.(ii) Assume that |α|−|β|=|α−β| holds for any two complex numbers α and β. AS α and β are complex, they can be represented as $\alpha =r_{1}e^{i\theta_{1}}$ and $\beta =r_{2}e^{i\theta_{2}}$ based on some non-negative real numbers $r_{1}$,$r_{2}$, $\theta_{1}$, and $\theta_{2}$. Without loss of generality, we may assume that $r_{2}\leq r_{1}$. Observe that $\left|\alpha \right|=r_{1}$, $\left|\beta \right|=r_{2}$, and
(15)$$\begin{array}{ccc} \left|\alpha -\beta \right| & = & |r_{1}e^{i\theta_{1}}-r_{2}e^{i\theta_{2}}\left|=\right|e^{i\theta_{1}}\left|\right|r_{1}-r_{2}e^{i(\theta_{2}-\theta_{1})}\left|=\right|\left(r_{1}-r_{2}\cos\left(\theta_{2}-\theta_{1}\right)\right)-r_{2}i\sin\left(\theta_{2}-\theta_{1}\right)|. \end{array}$$ Therefore, |α|−|β|=|α−β| implies that $\cos(\theta_{2}-\theta_{1})=1$; thus, $\theta_{2}=\theta_{1}$. Consequently, we have
(16)$$\beta =r_{2}e^{i\theta_{2}}=\frac{r_{2}}{r_{1}}\left(r_{1}e^{i\theta_{1}}\right)=p\alpha ,$$
where $p=r_{2}/r_{1}\geq 0$. For the inverse direction, assume that β=pα holds for some non-negative *p*. Since $r_{2}\leq r_{1}$, 1−p is non-negative. Thus, we have
(17)|α−β|=|α−pα|=|(1−p)α|=|1−p||α|=(1−p)|α|=|α|−p|α|=|α|−|β|,
which completes the proof. □

**Lemma** **2.**
*For any two vectors $|\eta \rangle$ and $|\zeta \rangle$ represented as*

(18)
$$|\eta \rangle =\sum_{j=0}^{d-1}c_{0j}|j\rangle \;\mathrm{and}\;|\zeta \rangle =\sum_{j=0}^{d-1}c_{1j}|j\rangle ,$$

*we have the equality*

(19)
$$\sum_{\begin{array}{c} j,l=0\\ j>l \end{array}}^{d-1}{\left|c_{0j}c_{1l}-c_{0l}c_{1j}\right|}^{2}=\left(\sum_{j=0}^{d-1}{\left|c_{0j}\right|}^{2}\right)\left(\sum_{j=0}^{d-1}{\left|c_{1j}\right|}^{2}\right)-\left(\sum_{j=0}^{d-1}c_{1j}^{*}c_{0j}\right)\left(\sum_{j=0}^{d-1}c_{0j}^{*}c_{1j}\right),$$

*where $c_{ij}$ are complex coefficients, and $|j\rangle$ indicates the computational basis of a d-dimensional Hilbert space.*


**Proof.** Consider the norm of the bipartite vector $|\eta \rangle ⊗|\zeta \rangle -|\zeta \rangle ⊗|\eta \rangle$, which is as follows:
(20)$$\begin{array}{ccc} {∥|\eta \rangle ⊗|\zeta \rangle -|\zeta \rangle ⊗|\eta \rangle ∥}^{2} & = & {∥\sum_{j=0}^{d-1}\sum_{l=0}^{d-1}\left(c_{0j}c_{1l}-c_{1j}c_{0l}\right)|j\rangle ⊗|l\rangle ∥}^{2} \end{array}$$
(21)$$\begin{array}{ccc} & = & \sum_{j=0}^{d-1}\sum_{l=0}^{d-1}{\left|c_{0j}c_{1l}-c_{1j}c_{0l}\right|}^{2} \end{array}$$
(22)$$\begin{array}{ccc} & = & \sum_{\begin{array}{c} j,l=0\\ j>l \end{array}}^{d-1}{\left|c_{0j}c_{1l}-c_{1j}c_{0l}\right|}^{2}+\sum_{j=1}^{d-1}{\left|c_{0j}c_{1j}-c_{1j}c_{0j}\right|}^{2}+\sum_{\begin{array}{c} j,l=0\\ j<l \end{array}}^{d-1}{\left|c_{0j}c_{1l}-c_{1j}c_{0l}\right|}^{2} \end{array}$$
(23)$$\begin{array}{ccc} & = & 2\sum_{\begin{array}{c} j,l=0\\ j>l \end{array}}^{d-1}{\left|c_{0j}c_{1l}-c_{1j}c_{0l}\right|}^{2}. \end{array}$$ In addition, the above quantity can be represented as
(24)$$\begin{array}{ccc} {∥|\eta \rangle ⊗|\zeta \rangle -|\zeta \rangle ⊗|\eta \rangle ∥}^{2} & = & \left(\langle \eta |⊗\langle \zeta |-\langle \zeta |⊗\langle \eta |\right)\left(|\eta \rangle ⊗|\zeta \rangle -|\zeta \rangle ⊗|\eta \rangle \right) \end{array}$$
(25)$$\begin{array}{ccc} & = & \langle \eta |\eta \rangle \langle \zeta |\zeta \rangle -\langle \eta |\zeta \rangle \langle \zeta |\eta \rangle -\langle \zeta |\eta \rangle \langle \eta |\zeta \rangle +\langle \zeta |\zeta \rangle \langle \eta |\eta \rangle \end{array}$$
(26)$$\begin{array}{ccc} & = & 2\left(\langle \eta |\eta \rangle \langle \zeta |\zeta \rangle -\langle \zeta |\eta \rangle \langle \eta |\zeta \rangle \right) \end{array}$$
(27)$$\begin{array}{ccc} & = & 2\left[\left(\sum_{j=0}^{d-1}{\left|c_{0j}\right|}^{2}\right)\left(\sum_{j=0}^{d-1}{\left|c_{1j}\right|}^{2}\right)-\left(\sum_{j=0}^{d-1}c_{1j}^{*}c_{0j}\right)\left(\sum_{j=0}^{d-1}c_{0j}^{*}c_{1j}\right)\right]. \end{array}$$ This completes the proof. □

## 3. Calculation of Global and Local Fidelities

In this section, we present the calculation of the global fidelity $F^{AB}$ and the local fidelity $F^{A}$ for any two pure quantum states ${|\psi \rangle }^{AB}$ and ${|\phi \rangle }^{AB}$. These calculations will be used in the next section.

Let us first consider the Schmidt decomposition [8] of the quantum state ${|\psi \rangle }^{AB}$, which is given by
(28)$${|\psi \rangle }^{AB}=\sqrt{\lambda }{|00\rangle }^{AB}+\sqrt{1-\lambda }{|11\rangle }^{AB}$$
for some λ∈[0,1/2]. In this equation, $\left\{{|0\rangle }^{A},{|1\rangle }^{A}\right\}$ and $\left\{{|0\rangle }^{B},{|1\rangle }^{B},\ldots ,{|d-1\rangle }^{B}\right\}$ are orthonormal bases on the quantum systems *A* and *B*, respectively. Then, the quantum state ${|\phi \rangle }^{AB}$ can be represented as
(29)$${|\phi \rangle }^{AB}=\sum_{i=0}^{1}\sum_{j=0}^{d-1}c_{ij}{|ij\rangle }^{AB},$$
where $c_{ij}$ are complex numbers satisfying
(30)$$\sum_{i=0}^{1}\sum_{j=0}^{d-1}{\left|c_{ij}\right|}^{2}=1.$$

Given that ${|\psi \rangle }^{AB}$ and ${|\phi \rangle }^{AB}$ are pure states, $F^{AB}$ can be calculated as
(31)$$\begin{array}{ccc} F^{AB} & = & {\left|{\langle \psi |}^{AB}{|\phi \rangle }^{AB}\right|}^{2} \end{array}$$
(32)$$\begin{array}{ccc} & = & {\left|\left(\sqrt{\lambda }{\langle 00|}^{AB}+\sqrt{1-\lambda }{\langle 11|}^{AB}\right)\left(\sum_{i=0}^{1}\sum_{j=0}^{d-1}c_{ij}{|ij\rangle }^{AB}\right)\right|}^{2} \end{array}$$
(33)$$\begin{array}{ccc} & = & {\left|\sqrt{\lambda }c_{00}+\sqrt{1-\lambda }c_{11}\right|}^{2}, \end{array}$$
where the second equality arises from Equations (Equation 28) and (Equation 29). In addition, the reduced states $\rho_{\psi }^{A}$ and $\rho_{\phi }^{A}$ of the quantum states ${|\psi \rangle }^{AB}$ and ${|\phi \rangle }^{AB}$ can be represented as
(34)$$\begin{array}{ccc} \rho_{\psi }^{A} & = & \lambda {|0\rangle }^{A}{\langle 0|}^{A}+\left(1-\lambda \right){|1\rangle }^{A}{\langle 1|}^{A}, \end{array}$$
(35)$$\begin{array}{ccc} \rho_{\phi }^{A} & = & \left(\sum_{j=0}^{d-1}{\left|c_{0j}\right|}^{2}\right){|0\rangle }^{A}{\langle 0|}^{A}+\left(\sum_{j=0}^{d-1}c_{1j}^{*}c_{0j}\right){|0\rangle }^{A}{\langle 1|}^{A}+\left(\sum_{j=0}^{d-1}c_{0j}^{*}c_{1j}\right){|1\rangle }^{A}{\langle 0|}^{A}+\left(\sum_{j=0}^{d-1}{\left|c_{1j}\right|}^{2}\right){|1\rangle }^{A}{\langle 1|}^{A}. \end{array}$$ Thus, the operator $\sqrt{\rho_{\psi }^{A}}\rho_{\phi }^{A}\sqrt{\rho_{\psi }^{A}}$ is represented as
(36)$$\begin{array}{ccc} \sqrt{\rho_{\psi }^{A}}\rho_{\phi }^{A}\sqrt{\rho_{\psi }^{A}} & = & \left(\sqrt{\lambda }{|0\rangle }^{A}{\langle 0|}^{A}+\sqrt{1-\lambda }{|1\rangle }^{A}{\langle 1|}^{A}\right)\rho_{\phi }^{A}\left(\sqrt{\lambda }{|0\rangle }^{A}{\langle 0|}^{A}+\sqrt{1-\lambda }{|1\rangle }^{A}{\langle 1|}^{A}\right) \end{array}$$
(37)$$\begin{array}{ccc} & = & \lambda {\langle 0|}^{A}\rho_{\phi }^{A}{|0\rangle }^{A}{|0\rangle }^{A}{\langle 0|}^{A}+\sqrt{\lambda (1-\lambda )}{\langle 0|}^{A}\rho_{\phi }^{A}{|1\rangle }^{A}{|0\rangle }^{A}{\langle 1|}^{A} \end{array}$$
(38)$$\begin{array}{ccc} & & +\sqrt{(1-\lambda )\lambda }{\langle 1|}^{A}\rho_{\phi }^{A}{|0\rangle }^{A}{|1\rangle }^{A}{\langle 0|}^{A}+\left(1-\lambda \right){\langle 1|}^{A}\rho_{\phi }^{A}{|1\rangle }^{A}{|1\rangle }^{A}{\langle 1|}^{A}. \end{array}$$ Consider an operator *L* defined as
(39)$$L=a_{00}{|0\rangle }^{A}{\langle 0|}^{A}+a_{01}{|0\rangle }^{A}{\langle 1|}^{A}+a_{10}{|1\rangle }^{A}{\langle 0|}^{A}+a_{11}{|1\rangle }^{A}{\langle 1|}^{A},$$
wherein the coefficients $a_{ij}$ are
(40)$$\begin{array}{ccc} a_{00} & = & \lambda {\langle 0|}^{A}\rho_{\phi }^{A}{|0\rangle }^{A}, \end{array}$$
(41)$$\begin{array}{ccc} a_{01} & = & \sqrt{\lambda (1-\lambda )}{\langle 0|}^{A}\rho_{\phi }^{A}{|1\rangle }^{A}, \end{array}$$
(42)$$\begin{array}{ccc} a_{10} & = & \sqrt{(1-\lambda )\lambda }{\langle 1|}^{A}\rho_{\phi }^{A}{|0\rangle }^{A}=a_{01}^{*}, \end{array}$$
(43)$$\begin{array}{ccc} a_{11} & = & \left(1-\lambda \right){\langle 1|}^{A}\rho_{\phi }^{A}{|1\rangle }^{A}. \end{array}$$ In addition, let us consider an operator *M* defined as
(44)$$M=b_{00}{|0\rangle }^{A}{\langle 0|}^{A}+b_{01}{|0\rangle }^{A}{\langle 1|}^{A}+b_{10}{|1\rangle }^{A}{\langle 0|}^{A}+b_{11}{|1\rangle }^{A}{\langle 1|}^{A},$$
wherein the coefficients $b_{ij}$ are
(45)$$\begin{array}{ccc} b_{00} & = & \frac{a_{00}}{a_{00}+a_{11}}, \end{array}$$
(46)$$\begin{array}{ccc} b_{01} & = & \frac{a_{01}}{a_{00}+a_{11}}, \end{array}$$
(47)$$\begin{array}{ccc} b_{10} & = & \frac{a_{10}}{a_{00}+a_{11}}=b_{01}^{*}, \end{array}$$
(48)$$\begin{array}{ccc} b_{11} & = & \frac{a_{11}}{a_{00}+a_{11}}. \end{array}$$ Then, *M* is positive, Hermitian, and has trace 1. Note that *L* and *M* satisfy the equality $L=(a_{00}+a_{11})M$.

Any operator *N*, expressed as
(49)$$N=a|0\rangle \langle 0|+b|0\rangle \langle 1|+b^{*}|1\rangle \langle 0|+\left(1-a\right)|1\rangle \langle 1|,$$
that is positive, Hermitian, and has trace 1, has eigenvalues $\lambda_{\pm }$ and eigenvectors $|\lambda_{\pm }\rangle$ given by
(50)$$\lambda_{\pm }=\frac{1\pm \sqrt{1-4a+4a^{2}+4{\left|b\right|}^{2}}}{2},$$
where a∈[0,1], b∈C, and $|0\rangle$ and $|1\rangle$ are orthonormal vectors. Note that $\mathrm{Tr}\left[N\right]=\lambda_{+}+\lambda_{-}=1$ and $\mathrm{Det}\left[N\right]=\lambda_{+}\lambda_{-}=a\left(1-a\right)-{\left|b\right|}^{2}$.

Consequently, the eigenvalues $\lambda_{1}$ and $\lambda_{2}$ of *M* are calculated as
(51)$$\begin{array}{ccc} \lambda_{1} & = & \frac{1+\sqrt{1-4b_{00}+4b_{00}^{2}+4{\left|b_{01}\right|}^{2}}}{2}, \end{array}$$
(52)$$\begin{array}{ccc} \lambda_{2} & = & \frac{1-\sqrt{1-4b_{00}+4b_{00}^{2}+4{\left|b_{01}\right|}^{2}}}{2}, \end{array}$$
and thus, the operator *L* has the eigenvalues $\left(a_{00}+a_{11}\right)\lambda_{1}$ and $\left(a_{00}+a_{11}\right)\lambda_{2}$. It follows that
(53)$$\mathrm{Tr}\sqrt{\sqrt{\rho_{\psi }^{A}}\rho_{\phi }^{A}\sqrt{\rho_{\psi }^{A}}}=\sqrt{\left(a_{00}+a_{11}\right)\lambda_{1}}+\sqrt{\left(a_{00}+a_{11}\right)\lambda_{2}}.$$ Since the trace and determinant of operator *M*, i.e., Tr[M]=1 and $\mathrm{Det}\left[M\right]=b_{00}b_{11}-{\left|b_{01}\right|}^{2}$, respectively, are known, we have
(54)$$\begin{array}{ccc} & & {\left(\mathrm{Tr}\sqrt{\sqrt{\rho_{\psi }^{A}}\rho_{\phi }^{A}\sqrt{\rho_{\psi }^{A}}}\right)}^{2} \end{array}$$
(55)$$\begin{array}{ccc} & & =\left(a_{00}+a_{11}\right)\left(\lambda_{1}+\lambda_{2}\right)+2\sqrt{{\left(a_{00}+a_{11}\right)}^{2}\lambda_{1}\lambda_{2}} \end{array}$$
(56)$$\begin{array}{ccc} & & =a_{00}+2\sqrt{{\left(a_{00}+a_{11}\right)}^{2}\left(b_{00}b_{11}-{\left|b_{01}\right|}^{2}\right)}+a_{11} \end{array}$$
(57)$$\begin{array}{ccc} & & =a_{00}+2\sqrt{\left(a_{00}a_{11}-{\left|a_{01}\right|}^{2}\right)}+a_{11} \end{array}$$
(58)$$\begin{array}{ccc} & & =\lambda {\langle 0|}^{A}\rho_{\phi }^{A}{|0\rangle }^{A}+2\sqrt{\lambda \left(1-\lambda \right)\left({\langle 0|}^{A}\rho_{\phi }^{A}{|0\rangle }^{A}{\langle 1|}^{A}\rho_{\phi }^{A}{|1\rangle }^{A}-{\left|{\langle 0|}^{A}\rho_{\phi }^{A}{|1\rangle }^{A}\right|}^{2}\right)}+\left(1-\lambda \right){\langle 1|}^{A}\rho_{\phi }^{A}{|1\rangle }^{A} \end{array}$$
(59)$$\begin{array}{ccc} & & =\lambda \sum_{j=0}^{d-1}|c_{0j}{|}^{2}+2\sqrt{\lambda \left(1-\lambda \right)\left(\left(\sum_{j=0}^{d-1}{\left|c_{0j}\right|}^{2}\right)\left(\sum_{j=0}^{d-1}{\left|c_{1j}\right|}^{2}\right)-\left(\sum_{j=0}^{d-1}c_{1j}^{*}c_{0j}\right)\left(\sum_{j=0}^{d-1}c_{0j}^{*}c_{1j}\right)\right)}+\left(1-\lambda \right)\sum_{j=0}^{d-1}{\left|c_{1j}\right|}^{2} \end{array}$$
(60)$$\begin{array}{ccc} & & =\lambda \sum_{j=0}^{d-1}|c_{0j}{|}^{2}+2\sqrt{\lambda (1-\lambda )}\sqrt{\sum_{\begin{array}{c} j,l=0\\ j>l \end{array}}^{d-1}{\left|c_{0j}c_{1l}-c_{0l}c_{1j}\right|}^{2}}+\left(1-\lambda \right)\sum_{j=0}^{d-1}{\left|c_{1j}\right|}^{2}, \end{array}$$
among which the last equality arises from Lemma 2 and the rest can be obtained from the definitions of the coefficients $a_{ij}$ and $b_{ij}$. Thus, the local fidelity $F^{A}$ is represented as
(61)$$F^{A}=\lambda \sum_{j=0}^{d-1}|c_{0j}{|}^{2}+2\sqrt{\lambda (1-\lambda )}\sqrt{\sum_{\begin{array}{c} j,l=0\\ j>l \end{array}}^{d-1}{\left|c_{0j}c_{1l}-c_{0l}c_{1j}\right|}^{2}}+\left(1-\lambda \right)\sum_{j=0}^{d-1}{\left|c_{1j}\right|}^{2}.$$

## 4. Necessary and Sufficient Conditions

In this section, we present our main result, which establishes the necessary and sufficient conditions for the fidelity equality, i.e., $F^{AB}=F^{A}$.

**Theorem** **1**(necessary and sufficient conditions). *Let ${|\psi \rangle }^{AB}$ and ${|\phi \rangle }^{AB}$ be pure quantum states on a bipartite quantum system AB such that dimA=2 and dimB=d≥2. The quantum states ${|\psi \rangle }^{AB}$ and ${|\phi \rangle }^{AB}$ satisfy the fidelity equality, i.e.,*
(62)$$F^{AB}=F^{A},$$
*if and only if they satisfy the following four conditions:*
(63)$$\begin{array}{ccc} \sqrt{\lambda }\left|c_{01}\right| & = & \sqrt{1-\lambda }\left|c_{10}\right|, \end{array}$$
(64)Re(c00c11*)=|c00c11|,
(65)$$\begin{array}{ccc} c_{ij} & = & 0,\;\forall j\geq 2, \end{array}$$
(66)$$\begin{array}{ccc} c_{01}c_{10} & = & pc_{00}c_{11}, \end{array}$$
*wherein the notations used are the same as those used in Equations (Equation 28) and (Equation 29), k is real, and p is real and non-negative.*

**Proof.** From Equation (Equation 11) of Lemma 1, it suffices to demonstrate that the fidelity equality $F^{AB}=F^{A}$ holds if and only if the two quantum states ${|\psi \rangle }^{AB}$ and ${|\phi \rangle }^{AB}$ meet Equations (Equation 63), (Equation 64), and (Equation 65) and the following condition:
(67)$$|c_{00}c_{11}\left|-\right|c_{01}c_{10}\left|\;=\;\right|c_{00}c_{11}-c_{01}c_{10}|.$$(i) Assume that the equality $F^{AB}=F^{A}$ holds. Then, Equations (Equation 33) and (Equation 61) imply the following equation:
(68)$${\left|\sqrt{\lambda }c_{00}+\sqrt{1-\lambda }c_{11}\right|}^{2}=\lambda \sum_{j=0}^{d-1}|c_{0j}{|}^{2}+2\sqrt{\lambda (1-\lambda )}\sqrt{\sum_{\begin{array}{c} j,l=0\\ j>l \end{array}}^{d-1}{\left|c_{0j}c_{1l}-c_{0l}c_{1j}\right|}^{2}}+\left(1-\lambda \right)\sum_{j=0}^{d-1}{\left|c_{1j}\right|}^{2}.$$ By applying the triangle inequality to the LHS, we obtain the following inequality:
(69)$$2\sqrt{\lambda (1-\lambda )}\left|c_{00}c_{11}\right|\geq \lambda |c_{01}{|}^{2}+2\sqrt{\lambda (1-\lambda )}\left||c_{00}c_{11}\left|-\right|c_{01}c_{10}|\right|+\left(1-\lambda \right){\left|c_{10}\right|}^{2}.$$ If $|c_{00}c_{11}\left|<\right|c_{01}c_{10}|$ holds, then the inequality in Equation (Equation 69) becomes
(70)$$4\sqrt{\lambda (1-\lambda )}\left|c_{00}c_{11}\right|\geq \lambda |c_{01}{|}^{2}+2\sqrt{\lambda (1-\lambda )}\left|c_{01}c_{10}\right|+\left(1-\lambda \right){\left|c_{10}\right|}^{2}.$$ By applying the inequality $|c_{00}c_{11}\left|<\right|c_{01}c_{10}|$ to Equation (Equation 70), we obtain
(71)$${\left(\sqrt{\lambda }|c_{01}|-\sqrt{1-\lambda }\left|c_{10}\right|\right)}^{2}<0,$$
which is a contradiction. Consequently, we have the inequality
(72)$$|c_{00}c_{11}\left|\geq \right|c_{01}c_{10}|.$$ By applying this inequality to Equation (Equation 69), we obtain the inequality
(73)$${\left(\sqrt{\lambda }|c_{01}|-\sqrt{1-\lambda }\left|c_{10}\right|\right)}^{2}\leq 0.$$ Thus, we have demonstrated that the equality $\sqrt{\lambda }|c_{01}|=\sqrt{1-\lambda }\left|c_{10}\right|$ holds, which is the same as the first sufficient condition given as Equation (Equation 63).Second, we note that the LHS of Equation (Equation 68) becomes
(74)$$\begin{array}{ccc} |\sqrt{\lambda }c_{00}+\sqrt{1-\lambda }c_{11}{|}^{2} & = & \left(\sqrt{\lambda }c_{00}+\sqrt{1-\lambda }c_{11}\right)\left(\sqrt{\lambda }c_{00}^{*}+\sqrt{1-\lambda }c_{11}^{*}\right) \end{array}$$
(75)$$\begin{array}{ccc} & = & \lambda |c_{00}{|}^{2}+\sqrt{\lambda (1-\lambda )}\left({\left(c_{00}c_{11}^{*}\right)}^{*}+c_{00}c_{11}^{*}\right)+\left(1-\lambda \right){\left|c_{11}\right|}^{2} \end{array}$$
(76)=λ|c00|2+2λ(1−λ)Re(c00c11*)+(1−λ)|c11|2. Therefore, the equality in Equation (Equation 68) becomes
(77)2λ(1−λ)Re(c00c11*)
(78)$$\begin{array}{ccc} & & =\lambda \sum_{j\neq 0}|c_{0j}{|}^{2}+2\sqrt{\lambda (1-\lambda )}\sqrt{\sum_{\begin{array}{c} j,l=0\\ j>l \end{array}}^{d-1}{\left|c_{0j}c_{1l}-c_{0l}c_{1j}\right|}^{2}}+\left(1-\lambda \right)\sum_{j\neq 1}{\left|c_{1j}\right|}^{2} \end{array}$$
(79)$$\begin{array}{ccc} & & \geq \lambda |c_{01}{|}^{2}+2\sqrt{\lambda (1-\lambda )}\left|c_{00}c_{11}-c_{01}c_{10}\right|+\left(1-\lambda \right){\left|c_{10}\right|}^{2} \end{array}$$
(80)$$\begin{array}{ccc} & & \geq \lambda |c_{01}{|}^{2}+2\sqrt{\lambda (1-\lambda )}\left||c_{00}c_{11}\left|-\right|c_{01}c_{10}|\right|+\left(1-\lambda \right){\left|c_{10}\right|}^{2} \end{array}$$
(81)$$\begin{array}{ccc} & & =2\sqrt{\lambda (1-\lambda )}\left|c_{00}c_{11}\right|. \end{array}$$ Here, the first inequality is obtained by eliminating a few of the non-negative terms, the second inequality arises from the reverse triangle inequality, and the last equality is obtained from the inequality in Equation (Equation 72) and the first sufficient condition Equation (Equation 63). This implies that Re(c00c11*)≥|c00c11| holds. Because any complex number *z* satisfies the inequality Re(z)≤|z|, we establish the second sufficient condition presented in Theorem 1.To obtain the third sufficient condition, presented as Equation (Equation 65), we use Equation (Equation 78) as follows:
(82)2λ(1−λ)Re(c00c11*)
(83)$$\begin{array}{ccc} & & =\lambda \sum_{j\neq 0}|c_{0j}{|}^{2}+2\sqrt{\lambda (1-\lambda )}\sqrt{\sum_{\begin{array}{c} j,l=0\\ j>l \end{array}}^{d-1}{\left|c_{0j}c_{1l}-c_{0l}c_{1j}\right|}^{2}}+\left(1-\lambda \right)\sum_{j\neq 1}{\left|c_{1j}\right|}^{2} \end{array}$$
(84)$$\begin{array}{ccc} & & \geq \lambda \sum_{j\neq 0}|c_{0j}{|}^{2}+2\sqrt{\lambda (1-\lambda )}\left||c_{00}c_{11}\left|-\right|c_{01}c_{10}|\right|+\left(1-\lambda \right)\sum_{j\neq 1}{\left|c_{1j}\right|}^{2} \end{array}$$
(85)$$\begin{array}{ccc} & & =\lambda |c_{01}{|}^{2}+\left(1-\lambda \right)|c_{10}{|}^{2}+\lambda \sum_{j\geq 2}|c_{0j}{|}^{2}+2\sqrt{\lambda (1-\lambda )}\left||c_{00}c_{11}\left|-\right|c_{01}c_{10}|\right|+\left(1-\lambda \right)\sum_{j\geq 2}{\left|c_{1j}\right|}^{2} \end{array}$$
(86)$$\begin{array}{ccc} & & =2\sqrt{\lambda (1-\lambda )}|c_{01}c_{01}|+\lambda \sum_{j\geq 2}|c_{0j}{|}^{2}+2\sqrt{\lambda (1-\lambda )}\left||c_{00}c_{11}\left|-\right|c_{01}c_{10}|\right|+\left(1-\lambda \right)\sum_{j\geq 2}{\left|c_{1j}\right|}^{2}, \end{array}$$
where the inequality is obtained by eliminating a few of the non-negative terms and applying the reverse triangle inequality and the last equality arises from the first sufficient condition given as Equation (Equation 63). From Equations (Equation 72) and (Equation 64), we have
(87)$$0\geq \lambda \sum_{j\geq 2}|c_{0j}{|}^{2}+\left(1-\lambda \right)\sum_{j\geq 2}{\left|c_{1j}\right|}^{2},$$
which yields the third sufficient condition given as Equation (Equation 65).By applying the first three conditions to the equality in Equation (Equation 68), we deduce the last condition given as Equation (Equation 67). This condition is equivalent to the fourth sufficient condition stated in Theorem 1, based on Equation (Equation 11) of Lemma 1.(ii) We assume the aforementioned four conditions to prove the converse of Theorem 1. Note that
(88)$$\begin{array}{ccc} F^{A} & = & \lambda \left(|c_{00}{|}^{2}+{\left|c_{01}\right|}^{2}\right)+2\sqrt{\lambda (1-\lambda )}\left|c_{00}c_{11}-c_{01}c_{10}\right|+\left(1-\lambda \right)\left(|c_{10}{|}^{2}+{\left|c_{11}\right|}^{2}\right) \end{array}$$
(89)$$\begin{array}{ccc} & = & \lambda |c_{00}{|}^{2}+2\sqrt{\lambda (1-\lambda )}\left|c_{00}c_{11}\right|+\left(1-\lambda \right){\left|c_{11}\right|}^{2} \end{array}$$
(90)=λ|c00|2+2λ(1−λ)Re(c00c11*)+(1−λ)|c11|2
(91)$$\begin{array}{ccc} & = & {\left|\sqrt{\lambda }c_{00}+\sqrt{1-\lambda }c_{11}\right|}^{2} \end{array}$$
(92)$$\begin{array}{ccc} & = & F^{AB}, \end{array}$$
where the first equality is obtained by applying the third necessary condition given as Equation (Equation 65) to the local fidelity $F^{A}$ given by Equation (Equation 61), the first and fourth conditions stated in Equations (Equation 63) and (Equation 67) lead to the second equality, and the third and fourth equalities arise from the second condition given as Equation (64) and from Equation (Equation 76), respectively. □

Theorem 1 implies the following corollary, which is nothing but the contrapositive of Theorem 1.

**Corollary** **1.**
*Let ${|\psi \rangle }^{AB}$ and ${|\phi \rangle }^{AB}$ be pure quantum states on a bipartite quantum system AB such that dimA=2 and dimB=d≥2. The quantum states ${|\psi \rangle }^{AB}$ and ${|\phi \rangle }^{AB}$ satisfy the fidelity inequality*

(93)
$$F^{AB}<F^{A}$$

*if and only if they fail to satisfy at least one of four necessary and sufficient conditions outlined in Theorem 1, where $F^{AB}$ and $F^{A}$ are defined in Equations (Equation 28) and (Equation 29), respectively.*


By employing Theorem 1 or Corollary 1, one can readily verify whether a pair of pure quantum states ${|\psi \rangle }^{AB}$ and ${|\phi \rangle }^{AB}$ satisfies the fidelity equality $F^{AB}=F^{A}$. As a special case of Theorem 1, if the quantum state ${|\psi \rangle }^{AB}$ is separable, then the four equivalence conditions are reduced to a single condition, as follows.

**Corollary** **2.**
*If ${|\psi \rangle }^{AB}$ is separable, then the fidelity equality $F^{AB}=F^{A}$ holds if and only if the following condition holds:*

(94)
$$c_{1j}=0,\;\forall j\neq 1,$$

*where $c_{ij}$ is defined in Equation (Equation 29).*


**Proof.** In Equation (Equation 28), if ${|\psi \rangle }^{AB}$ is separable, then λ=0, and thus, we have ${|\psi \rangle }^{AB}={|11\rangle }^{AB}$. Assuming that $F^{AB}=F^{A}$ holds, the first necessary and sufficient condition in Theorem 1 implies that $c_{10}=0$. Furthermore, from the third necessary and sufficient condition in Theorem 1, we have that $c_{1j}=0$ for any j≠1.For the inverse, let us assume that $c_{1j}=0$ holds for any j≠1. Note that for ${|\psi \rangle }^{AB}={|11\rangle }^{AB}$, the global fidelity $F^{AB}$ and the local fidelity $F^{A}$ are given by
(95)$$\begin{array}{ccc} F^{AB} & = & |c_{11}{|}^{2}, \end{array}$$
(96)$$\begin{array}{ccc} F^{A} & = & \sum_{j=0}^{d-1}{\left|c_{1j}\right|}^{2}, \end{array}$$
which implies that $F^{AB}=F^{A}$ because $c_{1j}=0$ for any j≠1. □

## 5. Representations for Fidelity Equality

Based on the primary results presented in Section 4, we provide specific forms of the quantum state ${|\phi \rangle }^{AB}$ when the quantum states ${|\psi \rangle }^{AB}$ and ${|\phi \rangle }^{AB}$ satisfy $F^{AB}=F^{A}$.

If ${|\psi \rangle }^{AB}$ is a separable state, denoted as ${|\psi \rangle }^{AB}={|11\rangle }^{AB}$, Corollary 2 implies that the other quantum state ${|\phi \rangle }^{AB}$ is represented as follows:(97)$${|\phi \rangle }^{AB}=c_{11}{|\psi \rangle }^{AB}+\sum_{j=0}^{d-1}c_{0j}{|0j\rangle }^{AB},$$
where $c_{1j}=0$ for any j≠1. This representation shows that ${|\phi \rangle }^{AB}$ is the linear combination of the orthogonal states ${|\psi \rangle }^{AB}$ and ${|0j\rangle }^{AB}$. Furthermore, these states are also orthogonal to each other in subsystem *A*. Specifically, when we consider subsystem *A*, ${|\psi \rangle }^{AB}$ and ${|0j\rangle }^{AB}$ become ${|1\rangle }^{A}$ and ${|0\rangle }^{A}$, respectively. Therefore, in this case, the quantum states ${|0j\rangle }^{AB}$ have no effects on the global and local fidelities, while ${|\psi \rangle }^{AB}$ and its coefficient $c_{11}$ determine them, i.e., $F^{AB}=\left|c_{11}\right|=F^{A}$.

On the contrary, let us consider the case that ${|\psi \rangle }^{AB}$ is entangled, i.e., λ∈(0,1/2] in Equation (Equation 28). Then, the third necessary and sufficient condition of Theorem 1 implies that
(98)$${|\phi \rangle }^{AB}=c_{00}{|00\rangle }^{AB}+c_{01}{|01\rangle }^{AB}+c_{10}{|10\rangle }^{AB}+c_{11}{|11\rangle }^{AB},$$
where $|c_{00}{|}^{2}+|c_{01}{|}^{2}+|c_{10}{|}^{2}+{\left|c_{11}\right|}^{2}=1$. From the first, second, and fourth conditions in Theorem 1, along with Lemma 1, the coefficients $c_{ij}$ have the following relations:(99)$$\begin{array}{ccc} c_{11} & = & kc_{00}, \end{array}$$(100)$$\begin{array}{ccc} c_{01} & = & r_{01}e^{i\theta_{01}}, \end{array}$$(101)$$\begin{array}{ccc} c_{10} & = & \frac{\sqrt{\lambda }}{\sqrt{1-\lambda }}r_{10}e^{i\theta_{10}}, \end{array}$$(102)$$\begin{array}{ccc} c_{00} & = & r_{01}\sqrt{\frac{\sqrt{\lambda }}{\sqrt{1-\lambda }}\frac{1}{pk}}e^{i(\theta_{01}+\theta_{10})/2}, \end{array}$$
where *k*, $\theta_{01}$, and $\theta_{10}$ are real numbers, and *p*, $r_{01}$, and $r_{10}$ are non-negative real numbers. Thus, the quantum state ${|\phi \rangle }^{AB}$ in Equation (Equation 98) becomes
(103)$${|\phi \rangle }^{AB}=c_{00}\left({|00\rangle }^{AB}+\sqrt{\frac{\sqrt{1-\lambda }}{\sqrt{\lambda }}pk}\alpha {|01\rangle }^{AB}+\sqrt{\frac{\sqrt{\lambda }}{\sqrt{1-\lambda }}pk}\alpha^{*}{|10\rangle }^{AB}+k{|11\rangle }^{AB}\right),$$
where the coefficient α is a complex number defined as $e^{i(\theta_{01}-\theta_{10})/2}$.

**Remark** **1.**
*The coefficient p in the representation of the quantum state ${|\phi \rangle }^{AB}$ in Equation (Equation 103) determines its entanglement properties. Specifically, ${|\phi \rangle }^{AB}$ given by Equation (Equation 98) is separable if and only if $c_{00}c_{11}=c_{01}c_{10}$ holds. Therefore, ${|\phi \rangle }^{AB}$ of Equation (Equation 103) is separable if and only if p=1. Consequently, for the case of p=1, the representation in Equation (Equation 103) simplifies to*

(104)
$${|\phi \rangle }^{AB}=c_{00}\left({|0\rangle }^{A}+\sqrt{\frac{\sqrt{\lambda }}{\sqrt{1-\lambda }}k}\alpha^{*}{|1\rangle }^{A}\right)⊗\left({|0\rangle }^{B}+\sqrt{\frac{\sqrt{1-\lambda }}{\sqrt{\lambda }}k}\alpha {|1\rangle }^{B}\right).$$



## 6. Conclusions

In this study, we have explored quantum fidelity and its fundamental properties. Specifically, we have focused on bipartite pure quantum states ${|\psi \rangle }^{AB}$ and ${|\phi \rangle }^{AB}$, where the dimension of quantum system *A* is two and the dimension of system *B* is arbitrary. We have introduced the global fidelity $F^{AB}$ and the local fidelity $F^{A}$ for these quantum states in Section 2. We have established the inequality $F^{AB}\leq F^{A}$ but the conditions under which these fidelities are equal remained unknown. In Section 4, we have provided the necessary and sufficient conditions for the fidelity equality $F^{AB}=F^{A}$. Additionally, in Section 5, we have presented specific representations of the quantum state ${|\phi \rangle }^{AB}$ when $F^{AB}=F^{A}$ is satisfied by ${|\psi \rangle }^{AB}$ and ${|\phi \rangle }^{AB}$.

In this study, our analysis was based on the assumption that the bipartite quantum states for calculating quantum fidelities are pure, and we have considered a fixed dimension of two for subsystem *A*. However, for future research, we propose investigating the necessary and sufficient conditions for fidelity equality in general bipartite states. Moreover, it would be valuable to explore the relationships between the amount of entanglement and fidelity equality, as quantum entanglement plays a crucial role in quantum communication tasks, although our current work does not focus on it. To the best of our knowledge, there is a lack of research addressing the connection between entanglement and fidelity equality. Therefore, elucidating these relationships would contribute significantly to the field. Additionally, we suggest examining a specific scenario in which one of our target states corresponds to the the isotropic state [10] or the Werner state [11].

## Data Availability

No new data were created or analyzed in this study. Data sharing is not applicable to this article.

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
