# Peer review of "The Necessary and Sufficient Conditions When Global and Local Fidelities Are Equal"

_entropy, 2023, doi:10.3390/e25071093_

Round 1
Reviewer 1 Report
The authors present the necessary and sufficient conditions when the global quantum fidelity in the bipartite system ABand the local quantum fidelity in the subsystem A are equal for two bipartite pure quantum states. There are also derivedspecific representations of quantum states satisfying the fidelity equality.
The article is clearly and well written, and the obtained results can present interest in applications for quantum information processing tasks with discrete variable systems.
I recommend the publication of this manuscript in the present form in the journal Entropy.
English language fine. No issues detected
Reviewer 2 Report
I have studied this work carefully and found that the Lemma 1 seems incorrect because how to prove (12) and (13) are equal to each other is not clear. If ad=bc, I cannot find (12) is equal to (13) mathematically.
it seems OK.
Reviewer 3 Report
The draft addresses a mathematical question to a well-established fact that the fidelity between two global quantum states is less or equal to the fidelity between reduced subsystems of the said states. The inequality is strictly inequal when the original states are highly entangled. However, the main question remains: what is the relation between the amount of entanglement and equality?
Present draft answers partially, for pure bipartite states, that equality to hold, one of the bipartite states should be close to a product state. I feel that there are certain avenues to improve the result:
1. It is clear to me that the entanglement of the second state \phi is close to zero. However, a proper analysis is needed to have an impactful result.
2. Can we say something about one pure state and the other one isotropic/Werner state? I hope this is possible to pursue.
3. Importance of this inequality is not discussed in the introduction. Some lines will improve its visibility.
Minor:
1. Line 77 has a typo. It should be \cos(\theta_2-\theta_1)=1.
2. Lemma 2 is misleading. It needs proper rephrasing. Including the vectors \eta and \xi will clarify it.
Reviewer 4 Report
In this work the authors studied an elementary property of quantum fidelity. Namely the fidelity between any two quantum states describing a bipartite system AB is less than or equal to the quantum fidelity between the reduced states in the subsystem A.
As a relevant results, they derived necessary and sufficient conditions when equality holds.
They also derived specific representations of quantum states satisfying the fidelity equality.
The subject is interesting and the results are scientifically sound.
However the paper is of light weight. In oder to be publishable, in my opinion, it should not be limited to the case of dimA = 2 (leaving for future only the analysis of mixed states).
As a minor concern, when introducing the notion of quantum fidelity the following papers should be cited for historical reasons:
R. Jozsa,, J. Mod. Opt. 41, 2315 (1994).
A. Uhlmann, Rep. Math. Phys. 9, 273 (1976)
Reviewer 5 Report
The authors derive sufficient and necessary conditions for equality of global and local fidelities of bipartite quantum states. The results reported by the authors are limited to pure bipartite states with dimension of one of the subsystems equal to 2. Nevertheless, already for this simplest instance the problem becomes nontrivial. The authors discuss and analyze the obtained equality conditions which provides some insight into their structure. The results are new and deserve to be published.
Round 2
Reviewer 3 Report
I am happy with the revision and recommend it for publication.
Reviewer 4 Report
The authors did not address my main concern, i.e. the fact that the paper is of light weight and the analysis should not be limited to the case of dimA = 2. They even did not address the case of dimA = 3, claiming that is not easy. I agree that the extension to dim A > 2 could not be easy, but the relevance of a publication is also measured by the difficulty of the addressed problem. If I do not see at least an extension to the case of dimA = 3, I cannot recommend publication in this journal (eventually the paper could be suitable for a minor journal).